# The Effect of Monophasic Pulsed Current with Stretching Exercise on the Heel Pain and Plantar Fascia Thickness in Plantar Fasciitis: A Randomized Controlled Trial

**DOI:** 10.3390/healthcare8020079

**Published:** 2020-03-30

**Authors:** Abdullah Alotaibi, Jerrold Petrofsky, Noha S. Daher, Everett Lohman, Hasan M. Syed, Haneul Lee

**Affiliations:** 1Department of Physical Therapy, Loma Linda University, Loma Linda, CA 92354, USA; alotaibiak@psmmc.med.sa (A.A.); elohman@llu.edu (E.L.); 2Scientific Research Center, Saudi Armed Forces Medical Service Department, Riyadh 11461, Saudi Arabia; 3School of Physical Therapy, Touro University, Henderson, NV 89014, USA; Jerrold.Petrofsky@tun.touro.edu; 4School of Allied Health Professions, Loma Linda University, Loma Linda, CA 92354, USA; ndaher@llu.edu; 5Department of Orthopedic Surgery, School of Medicine, Loma Linda University, Loma Linda, CA 92354, USA; hsyed@llu.edu; 6Department of Physical Therapy, Gachon University, Incheon 21936, Korea

**Keywords:** plantar fasciitis, plantar fascia thickness, ultrasound, monophasic pulsed current, stretching exercises

## Abstract

Plantar fasciitis (PF) is one of the most common causes of heel and foot pain. Monophasic pulsed current (MPC) is an electrical stimulation used to accelerate the healing processes. The purpose of this study was to determine the effect of MPC and MPC combined with plantar fascia stretching exercises (SE) on heel pain and plantar fascia thickness in treatment of PF and see if there is any relationship between heel pain and plantar fascia thickness after intervention. Forty-four participants diagnosed with PF were randomly assigned to two group; MPC group or MPC combined with plantar fascia SE. Plantar fascia thickness was measured with musculoskeletal ultrasound. Although no statistical differences between the two groups were found, heel pain and the plantar fascia thickness significantly decreased in both groups after the intervention (*p* < 0.001). No significant correlation was found between changes in heel pain and plantar fascia thickness after 4 weeks of treatment. Our results indicated that MPC can reduce heel pain and plantar fascia thickness caused by PF. However, MPC combined with plantar fascia SE is not superior to MCP only in terms of reduction in heel pain and plantar fascia thickening.

## 1. Introduction

Plantar fasciitis (PF) is one of the most common causes of heel and foot pain and it was first described in 1812 [1,2]. Proximal PF is a clinical diagnosis usually affecting more than two million Americans every year [2,3]. Among then, more than one million visit the physician or foot specialists for pain management per year and the annual cost of this treatment is estimated 284 million US dollars [2,4].

Plantar fascia or plantar aponeurosis is a thick and strong fibrous connective tissue extending from the medial tuberosity of the calcaneus and into three bands to attach into the bases of proximal phalanges or at the metatarsophalangeal joints to the medial longitudinal arch of the foot [5,6].

Extrinsic potential predisposing factors that may make someone susceptible to the development of PF include high intensity sport activities or training that require repetitive plantar flexion and extension of the metatarsophalangeal, and that mechanical overload and high tensile load that develop micro-tears of the plantar fascia, leading chronic inflammatory responses followed by degeneration [7]. Other extrinsic potential risk factors include the use of poor or worn footwear, occupational and recreational activities that require prolonged standing or weight bearing, and improper posture [5,8,9].

The classic feature and the presentation of PF are mechanical symptoms of pain on the plantar foot at the end of the heel [9,10]. Pain may interfere with walking, particularly when taking the first few steps in the morning after exiting the bed, or arising from a seat after prolonged sitting or inactivity. The diagnosis of PF used to be made through a thorough and comprehensive history and physical examination. Subjective heel pain during the first few steps in the morning is a typical symptom of PF, which is distinguished from other heel pain.

In recent years, musculoskeletal ultrasound (MSK US) has been widely used as a diagnostic tool to corroborate or verify a clinical diagnostic entity of PF [11,12]. Numerous diagnostic sonography studies showed that abnormal thickening of plantar fascia greater than 4 mm and reduced echogenicity are generally related to PF [13,14,15]. Many clinical trials have been conducted to measure the plantar fascia thickness to prove that treatment’s efficacy [11,14,15].

A patient with PF is commonly instructed to have rest and avoid any strenuous and arduous activities that place strain on the inflamed and irritated proximal insertion of plantar fascia [3,16,17]. It is well known that physical therapy has a positive effect on treatment of PF in terms of reliving heel pain [9]. Many physical therapy treatment options are available which may attenuate the heel pain caused by PF [9,18]. Physical therapy modalities include iontophoresis, icing, contrast baths, ultrasound, taping, night splinting, and customized foot sole insets, shoe modification which can be used for patient needs [3,9,16,17]. Other physical therapy techniques include soft tissue mobilization, manual therapy, and plantar fascia stretching exercises (SE) [19,20]. Plantar fascia SE, an integral component of the therapeutic exercise as a treatment for reducing pain and functional limitations, has been widely prescribed as a conventional therapy to reduce tension in the foot [3]. In addition, rest and avoiding vigorous activities that place strain on the inflamed and irritated proximal insertion of plantar fascia may also ally inferior heel symptoms [3,16]

Fibroblasts, synthesizing the collagens, elastin fibers, and glycoproteins found in the extracellular matrix to produce the structural framework, play a key role during the proliferation phase of the healing process [21,22]. Monophasic pulsed current (MPC) promotes wound healing processes with using the negative electrode to attract the fibroblast cells to promote and accelerate healing, especially, the proliferation phase. With electrodes, it delivers electrical currents directly to the wound bed and appears to increase cellular actions and histological responses such as collagen synthesis, producing adenosine triphosphate, increasing the number of growth factor receptor and calcium influx [21,22,23]. Other than these cellular actions, MPC improves tissue perfusion and decreased edema [24].

The primary purpose of this study was to examine the effect of MPC and MPC combined with plantar fascia SE on the heel pain and plantar fascia thickness on patients diagnosed with PF. The secondary object was to investigate the correlation between changes of heel pain and plantar fascia thickness after completing 4-week treatment of MPC and MPC with SE. 

## 2. Materials and Methods

### 2.1. Ethical Approval

This study was approved by the Institutional Review Board (IRB) at Loma Linda University (LLU IRB #5130018). All participants received explanations about the procedures and signed a statement of informed consent prior to participation in the study.

### 2.2. Subjects

Individuals with PF diagnosed with radiologic proof of heel spur (either US or MRI) were recruited from community-based referring physicians. Subjects were included if they have a primary clinical diagnosis of PF determined by tenderness to pressure on the medial tubercle of the calcaneus, as well as a complaint of heel pain associated with first step after walking in the morning greater than 3 out of 10 on a visual analogue scale (VAS) at least three months. Patients were excluded if they had any fractures or surgeries of the lower limbs or any specific metabolic and connective tissue disorders related to the diagnosis of PF. Patients with any allergy to electrode/tape/gel and contraindications for MPC such as patients with pacemakers, recent hemorrhage, open wounds, or compromised circulation were also excluded from the study.

### 2.3. Procedure

Once patients singed the informed consent, the researcher obtained a demographics of the subjects including age, gender, height, weight, body mass index (BMI), average standing hours per day, and duration of the symptoms. Then, subjective heel pain using VAS and sagittal thickness of proximal plantar fascia with MSK US were measured for baseline measurement.

Then, they were randomly assigned into two groups; either MPC group or MPC combined with plantar fascia SE group. Simple randomization methods with a computer-generated random two-digit number was performed before the data collection.

Both groups received 3 sessions of MPC (60-min per session) per week for 4 weeks at the clinic. Additionally, patients in MPC combined with plantar fascia SE group had an extra session of plantar fascia SE when they had MPC treatment for the first time and were provided a protocol of the home-based SE program. Measurements of subjective heel pain and plantar fascia thickness were conducted after 4 weeks of intervention in both groups. 

### 2.4. Outcome Measures

#### 2.4.1. Subjective Pain 

The VAS was used to measure subjective heel pain. Subjects were asked to place a vertical mark across a VAS chart and rate their subjective heel pain of an initial steps in the morning. VAS is the most used pain scale with marked point ranges from 0 to 10, with 0 indicating no pain, and 10 indicating the extreme pain [25]. 

#### 2.4.2. Plantar Fascia Sagittal Thickness

A Mindray-M7 Diagnostic Ultrasound System with L14-6 MHz linear probe (Mindray Bio-Medical Electronics, Shenzhen, China) was used for evaluating plantar fascia sagittal thickness. The plantar fascia is most effectively assessed with the foot hanging over the edge at the table with neutral position of ankle in the prone position. The ultrasound probe was placed vertically in relation to the plantar aspect of the heel. The sagittal plantar fascia thickness was measured at the medial calcaneal tubercle insertion with 5 mm distal from the medial calcaneal tuberosity. All plantar fascia thickness measurements were taken by the 5 more year experienced and certified Musculoskeletal sonographer.

The standard normal or asymptomatic plantar fascia thickness is range from 2.3 to 4.0 mm [14,15] and greater than 4 mm would be considered as the presentation of PF [26,27].

### 2.5. Interventions

#### 2.5.1. Monophasic Pulsed Current

The GV 350 Galvanic High-Volt Pulsed Stimulator (Biomedical Life System, Carlsbad, CA, USA) was used. The MPC is a percutaneous delivery of monophasic pulsed current, twin-peak, pulses with phase duration between 50 and 150μ second (average 100 μ second), which employs voltage up to 500 volts [21,23]. For accelerating proliferation, it is used the negatively charged cathode to attract fibroblast cells due to polarity selection is based on the healing process the practitioner to facilitate and accelerate proliferation. The therapeutic parameters used in the study included: pulsed current, twin peaked, negative electrode polarity cathode, 100 μ second as frequency, 10,000 μ second for pulse duration, and at sub-motor level amplitude (Figure 1).

#### 2.5.2. Plantar Fascia Stretching Exercise

Subjects in MPC combined with SE group were instructed to cross the affected leg over the other leg on sitting, and holding the foot with their opposite hand, apply metatarso-phalangeal joint dorsiflexion while holding each stretch for 10 s, and repeating each stretch 10 times [3]. Patients in MPC combined with SE group were asked to perform the SE 3 times a day and the first SE should be done before exiting the bed (Appendix A). They were allowed using a towel to stretch the foot if they had a difficulty in holding otherwise. A PF SE log was given to subjects to record their home exercise compliance (Appendix A) and the SE compliances were 92.4 ± 2.8 %. 

### 2.6. Sample Size Estimation

SAS (Statistical Analysis System) statistical analysis software was used to calculate the sample size required for the study. A moderate expected effect size of 0.25 for repeated measures time and group interaction was applied [28], with an alpha error probability of 0.05 and a power of 0.85. A sample size of 38 was required to show statistical significance when clinically significant and additional subjects were recruited to provide for unanticipated attrition. 

### 2.7. Data Analysis

IBM SPSS 22.0 software was used to analyze the data. Data were summarized as means and SDs for continuous variables and frequencies and percentages for categorical variables to determine if significant differences between the two the groups existed. The assumption of normality of the continuous variables was examined using the Shapiro–Wilk test and all outcome variables were normally distributed. The two groups were compared at baseline using independent t-test. An analysis of variance (ANOVA) with repeated measures was conducted to examine an interaction in heel pain and plantar fascia thickness between the two groups over time. Pearson’s correlation was conducted to see if there is any correlation between the reduction in heel pain and plantar fascia thickness. The level of significance was set at α = 0.05.

## 3. Result

Forty-four subjects completed the study. The general characteristics are described in Table 1.

At baseline evaluation, there was no statistically significant difference in VAS between the MPC group and MPC combined with SE group (*p* > 0.05). Both groups experienced improvements after 4 weeks of treatment compared with heel pain at baseline (*p* < 0.01). The MPC group had decrease in heel pain of 3.96 scores (95% confidence interval (CI), 3.10 to 4.81) compared to mean reduction of 3.30 (95% CI, 2.40 to 4.91) for MPC combined with SE group after the intervention. There was no significant difference in mean heel pain between the two groups with mean difference of 0.11 (95% CI, 1.07 to 1.30; Table 2).

Similarly, no significant difference in plantar fascia thickness existed between the MPC group and MPC combined with SE group at baseline evaluation (*p* = 0.14). The two groups experienced significant reduction in the sagittal thickness of plantar fascia after 4-week treatment compared with baseline evaluation (*p* < 0.01), but difference between the two groups was not statistically significant (*p* = 0.23, Table 2). After treatment, the MPC group had a mean decrease in plantar fascia thickness of 0.74 mm (95% Cl, 0.55 to 0.93 mm) compared to mean reduction of 0.66 mm (95% CI, 0.51 to 0.80 mm) for patients in MPC combined with SE group (Table 2).

After treatment, the average reduction in heel pain was 3.63 ± 1.98 and the average reduction in the plantar fascia thickness was 0.37 mm ± 0.07 mm. There was no significant correlation between the mean changes in heel pain and the plantar fascia thickness measurement between pre- and post 4-week treatment (r = −0.006, *p* = 0.97; Figure 2). 

## 4. Discussion

PF is one of the most common musculoskeletal conditions seen in outpatient orthopedic settings [1,2]. It is associated with morning inferior heel pain especially when taking the first few steps upon rising [1,2,29] and it is further linked to abnormal thickening of the proximal plantar fascia [8,30]. 

The primary purpose of this study was to examine the effect of MPC and MPC combined with plantar fascia SE on heel pain and plantar fascia thickness in a treatment of PF. Our hypothesis was that the MPC would accelerate the plantar fascia healing process in terms of reduction in heel pain and fascia thickening from inflammation of a thick band of tissue. Plantar fascia is a connective tissue and the fibroblast cells main role is to stabilize the foot structure. The promotion and acceleration of healing processes of the inflamed proximal plantar fascia may decrease heel pain, tenderness, improve functional activities, and reduce abnormal thickening of plantar fascia associated with PF. 

This study revealed a significant decrease in the heel pain and plantar fascia thickness after the use of MPC and MPC combined with plantar fascia SE while two groups were insignificant in terms of the two outcome measures. Findings of this study agreed with previous studies about the efficacy of medical treatment options in reducing abnormal proximal thickening of planter fascia caused by PF [8,12,13,27,30,31]. Plantar fascia is a connective tissue and it is composed of fibroblast cell which make the collagens, glycosaminoglycans, elastin fibers, and glycoproteins [21,22]. Since polarity selection is based on the healing phase, negatively charged cathode was used to attract the positively charged fibroblast cells to promote and accelerate proliferation phase of plantar fascia in the present study which may help decrease plantar fascia thickness.

We applied plantar fascia SE in our study because it is considered central to most conservative treatment protocols and common exercise techniques performed easily by patients for inferior heel symptoms due to PF, especially in reducing heel pain [3]. Digiovanni BF et al. reported that plantar fascia specific-stretching protocol decreased pain and functional limitation for patients with chronic PF [3]. Similarly, Landorf KB et al. reported a significant improvement in heel pain and functional mobility in patients diagnosed with PF after using either custom or prefabricated foot orthoses for three months [17].

The current study further sought to examine whether a statistically significant correlation existed between changes in plantar fascia thickness and changes in inferior heel pain while simultaneously evaluating the effectiveness of MPC and MPC combined with plantar fascia SE on the treatment of PF. However, no significant relationship was observed, even though this study demonstrated a significant statistical decrease in heel pain and also in the plantar fascia thickness after 4-week intervention. In other words, receiving MPC itself is far more efficient in terms of time management. 

However, the following limitations should be addressed. The investigator was not blinded to group assignment and outcome measurements, which may bias the results of the study. Secondly, patients with symptoms less than 12 months would be needed to draw sound inferences about the effect of MPC. Third, study should include control or a placebo MPC group for better comparison. In addition, even though PF thickness was measured by an experienced sonographer, reliability was not tested. Further study should have the reliability of the researcher for more consistent and accurate results.

Shear-wave ultrasound elastography has been used for early diagnosis of PF and the measurement of elasticity of the plantar fascia may add a new dimension to the study since elasticity may be altered independent of plantar thickness [32,33]. Another useful technique is photoacoustic ultrasound imaging. This technique may allow better imaging of the plantar fascia since it is a superficial tissue and may provide more information as to damage and inflammation. It can provide molecular information absent from US images and show resolution in damage not detectable by simple ultrasound measurements [34].

## 5. Conclusions

In conclusion, MPC combined with plantar fascia SE is not superior to MPC only to decrease the heel pain and the plantar fascia thickness. Although, both MPC and MPC combined with plantar fascia SE showed significant decreases in heel pain and plantar fascia thickness caused by PF. Additionally, no significant relationship existed between the change between pre and post intervention in plantar fascia thickness and heel pain when evaluating the effect of MPC on the treatment PF.

Applying 12-session of MPC results in significant reduction of heel pain and sagittal thickness of plantar fascia and that this treatment can be applied in the clinical setting since many of the electrical stimulation units include the MPC mode that we used in our study.

## Figures and Tables

**Figure 1 healthcare-08-00079-f001:**
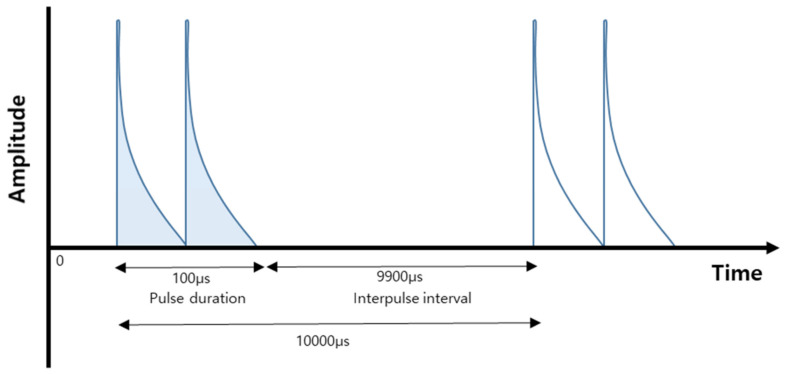
Illustration of the monophasic pulsed current with twin-peak.

**Figure 2 healthcare-08-00079-f002:**
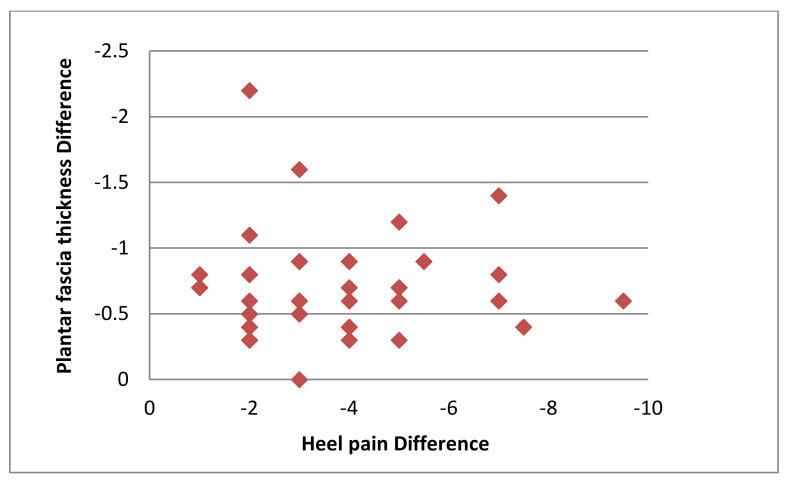
Scatter plot of the relationship between the reduction in heel pain and plantar fascia thickness.

**Table 1 healthcare-08-00079-t001:** General characteristics of subjects (*n* = 44).

Characteristic		MPC Group (*n* = 22)	MPC + SE Group (*n* = 22)	*p*-Value
Age, year		49.7 ± 11.7	49.0 ± 9.7	0.60
Height, cm		171.5 ± 12.0	171.0 ± 13.5	0.91
Weight, kg		96.4 ± 22.9	87.4 ± 22.9	0.20
BMI, kg/m^2^		32.8 ± 7.2	30.0 ± 7.4	0.21
Standing hours per day, hour		8.8 ± 3.2	9.6 ± 2.48	0.31
Duration of symptom, month		12.5 ± 1.2	12.9 ± 1.8	0.12
Gender	Female, % (*n*)	63.6 (14)	68.2% (15)	0.75
Involved side	RT, % (*n*)	27.3 (6)	50.0 (11)	0.12

Abbreviations: MPC, Monophasic pulsed current; SE, Stretching exercise; BMI, Body mass index; RT, Right.

**Table 2 healthcare-08-00079-t002:** Heel pain and plantar fascia thickness by treatment group over time.

Outcome Variables	Baseline (Mean ± SD)	Post-Intervention (Mean ± SD)	*p*-Value *	*p*-Value ^#^	Pre-Post by-Group Interaction
Heel pain					
MPC (*n* = 22)	7.39 ± 1.75	3.43 ± 1.95	< 0.01	0.67	0.28
MPC+SE group (*n* = 22)	6.84 ± 2.14	3.55 ± 1.95			
Plantar fascia thickness					
MPC (*n* = 22)	4.61 ± 1.19	3.87 ± 1.19	< 0.01	0.23	0.49
MPC+SE group (*n* = 22)	4.11 ± 0.99	3.45 ± 1.06			

Abbreviations: MPC, Monophasic pulsed current; SE, Stretching exercise; * Significant differences between baseline and post-intervention between two groups; ^#^ Significant differences between two groups at post-intervention.

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
