# Peer review of "The Effect of Monophasic Pulsed Current with Stretching Exercise on the Heel Pain and Plantar Fascia Thickness in Plantar Fasciitis: A Randomized Controlled Trial"

_healthcare, 2020, doi:10.3390/healthcare8020079_

Round 1

Reviewer 1 Report

The authors have tried to correlate the effect of monophased pulsed current and the stretching exercise on the thickness of Plantar fasciitis and the pain level in subjects. Below are the concerns that may need to be addressed before.

  1. How did the authors control or make sure that the stretching excercise were performed regularly by each subjects?
  2. The MPC+SE group had more standing hours per day. Does the MPC+SE group represent higher risk because of long hours of standing?
  3. The authors should also mention that the measurement can be made better by combining with photoacoustic contrast and cite the following literature (IEEE Trans Ultrason Ferroelectr Freq Control. 2015 Feb; 62(2): 319–328; Photoacoustics, Volume 9, 2018, Pages 10-20)    
  4. The plot of electric signal used in the MPC will be helpful(Amplitude Vs time)

Author Response

Dear Reviewer,

We are greateful for your suggestions. The manuscript has been revised according to all the required recommendations.

  1. How did the authors control or make sure that the stretching exercise were performed regularly by each subjects?
  • Patients were provided a written protocol of the stretching program and asked to keep a daily log of exercise completion for 4 weeks (APPENDIX A and B) and exercise compliance was 92.4 ± 2.8 %. We added this in the text. [Line155-157]
  1. The MPC+SE group had more standing hours per day. Does the MPC+SE group represent higher risk because of long hours of standing?
  • Mean standing hours per day in MPC+SE groups is more (0.8 hour = around 48 mins) compared to MPC group but it was not statistically different. (Even cohen’d is 0.279 which is small effect size) [Line 176-177, Table 1]
  1. The authors should also mention that the measurement can be made better by combining with photoacoustic contrast and cite the following literature (IEEE Trans Ultrason Ferroelectr Freq Control. 2015 Feb; 62(2): 319–328; Photoacoustics, Volume 9, 2018, Pages 10-20)    
  • We mentioned it and added the reference at the end of discussion. [Line 251-255]
  1. The plot of electric signal used in the MPC will be helpful (Amplitude Vs time)
  • Figure 1 has been added [Line 147-149]

Reviewer 2 Report

Dear authors:
It has been a pleasure to review your paper about “The Effect of Monophasic Pulsed Current with Stretching Exercise on the Heel Pain and Plantar Fascia Thickness in Plantar Fasciitis” but I have observed several of methodology errors that it’s necessary to change it

You can see below the recommendation

Title: Can you include the type of research in the title?

Introduction

The first paragraphs are very repetitive, more or less say the same in both texts. Can you improve this?

Line 43 Which is the differences between hell and rearfoot? Please can you clarify this?

In section method

You should include in the paragraph of ethical information the name and number of ethical committee and did you register the RCT? Can you include the number of registration?

Line 98-100 You write during the paper that you used US but why did you use this as an inclusion criteria? I think that it’s more real than the VAS pain by pressure, the pain with this test could be another thing, can you clarify and reference why?

Line 105-116 You explain at the end of the paper that the clinicians were not blind, but you can include this in the procedure too, it’s a big limitation of the research

Line 119-122 Can you include references for this? It’s very important that the tools are supported with references

Line 124-130 Did you do the reliability of the sonographer? It is not only necessary to say that he has experienced, but the paper should also have the reliability of the researcher

How did you calculate the sample size, can you include this in the text?

In general, the protocol is a little bit confused to read and can you improve?

In the statistical analysis, the sample is less than 50 participants, why did you use the Shapiro Will and what variable were normal and which no?

Discussion

Line 210-215 This paragraph is very simple, can you improve it? You lost information about the references and you can compare more with your data

I can not see the clinical application please can you include?

Author Response

Dear Reviewer,

Thank you so much for your suggestion. The manuscript has been extensively revised according to all the required recommendations.

Title: Can you include the type of research in the title?

  • We revised the title “The effect of Monophasic Pulsed Current with Stretching Exercise on the Heel Pain and Plantar Fascia Thickness in Plantar Fasciitis: A Randomized Controlled Trial” [Line 4-5]

Introduction

The first paragraphs are very repetitive, more or less say the same in both texts. Can you improve this? Line 43 Which is the differences between hell and rear foot? Please can you clarify this?

  • Revised [Line 43-46]

In section method

You should include in the paragraph of ethical information the name and number of ethical committee and did you register the RCT? Can you include the number of registration?

  • We have approved from Loma Linda University IRB, and we added approval number [Line 96] and unfortunately, we did not register the clinical trial number for this study.

Line 98-100 You write during the paper that you used US but why did you use this as an inclusion criterion? I think that it’s more real than the VAS pain by pressure, the pain with this test could be another thing, can you clarify and reference why?

  • I agree with you that Musculoskeletal ultrasound (MSK US) is a valid and valuable diagnostic imaging tool utilized for confirming the diagnosis of plantar fasciitis, we had subjects referred from local orthopedic clinics diagnosed by physician. Physicians made a diagnosis based on either US or MRI. We added this under methods. [Line 100-104]
    But still other studies used the same criterion that we used in our study as a provision for the participation eligibility. 

Line 105-116 You explain at the end of the paper that the clinicians were not blind, but you can include this in the procedure too, it’s a big limitation of the research

  • We have mentioned in the study that the assessor (not the clinician) was not blinded to treatment allocation and outcome assessment, and yes it is a potential source of bias. To overcome that sort of bias, the investigator performed the baseline evaluation for the measure the heel pain using VAS (subjective self-reported outcome measure) and the measurement of the sagittal thickness of the plantar fascia was evaluated by the investigator who is trained in MSK US (objective outcome measure). Then the investigator randomly assigned the participants to one of two treatment groups. The treatment was performed by clinicians under the supervision of one of the investigators who is orthopedic certified specialist. After treatment the post treatment evaluation was performed [Line 110-113]

Line 119-122 Can you include references for this? It’s very important that the tools are supported with references

  • Reference has been added [Line 127]

Line 124-130 Did you do the reliability of the sonographer? It is not only necessary to say that he has experienced, but the paper should also have the reliability of the researcher

  • We could not measure test-retest, inter- or intra reliability
  • Many of previous study which determined the effect of the specific treatment on PF thickness used similar protocols [Line 134-135] but we respect your previous opinion and we add this in the limitation [Line 246-248].
    1) Wu Ch et al. Sonoelastographic evaluation of plantar fascia after shock wave therapy for recalcitrant plantar fasciitis: A 12-month longitudinal follow-up study. Sci Rep. 2020 13;10:2571

How did you calculate the sample size, can you include this in the text?

  • Sample size estimation has been added under methods [Line 158-163]

In general, the protocol is a little bit confused to read and can you improve?

  • Protocol has been revised [Line 110-121]

In the statistical analysis, the sample is less than 50 participants, why did you use the Shapiro Will and what variable were normal and which no?

  • K-S is more general but you are right, Shapiro-wilk test is more suitable since it is more sensitive. We conducted a Shapiro wilk test and all variables were normally contributed.
    We changed the test name and added sentence [Line 168-169].

Discussion

Line 215-220 This paragraph is very simple; can you improve it? You lost information about the references and you can compare more with your data

  • Discussion has been revised [Line 220-234]

I cannot see the clinical application please can you include?

  • Clinical application has been added at the end of conclusion [Line 262-264]  

Round 2

Reviewer 2 Report

The changes have been done correctly and can be accepted in this form